# Consumers’ Food Safety Risk Communication on Social Media Following the Suan Tang Zi Accident: An Extended Protection Motivation Theory Perspective

**DOI:** 10.3390/ijerph18158080

**Published:** 2021-07-30

**Authors:** Ying Zhu, Xiaowei Wen, May Chu, Gongliang Zhang, Xuefan Liu

**Affiliations:** 1College of Economics & Management, South China Agricultural University, Guangzhou 510642, China; zhuyingsom@stu.scau.edu.cn (Y.Z.); xfliu@stu.scau.edu.cn (X.L.); 2Research Institute of Rural Development of Guangdong Province, Guangzhou 510642, China; 3Department of Government and Public Administration, United College, Chinese University of Hong Kong, Hong Kong 999077, China; maychu@cuhk.edu.hk; 4Management College, Zhongkai University of Agriculture and Engineering, Guangzhou 510408, China; zhangdry1984@163.com

**Keywords:** food risk communication, protection motivation theory, social media, SEM

## Abstract

There are many hidden safety hazards in homemade food due to an absence of food preparation and storage knowledge, and this has led to many food safety incidents. The purpose of this study was to explore the influencing factors of consumers’ food risk communication behavior on social media in northeast China, using the protection motivation theory. We integrate the Suan Tang Zi food poisoning accident and the protection motivation theory to develop a conceptual model to predict food safety risk communication on social media. We conducted a questionnaire which adapted measures from the existing Likert scales. A total of 789 respondents from northeast China participated in this study. We tested our hypotheses using a structural equation model. Results show that perceived severity, perceived vulnerability and self-efficacy have a significant influence on consumer protection motivation. Response efficacies have a positive impact on consumer protection motivation, but response barriers have a negative impact on consumer protection motivation. Additionally, information need and protection motivation of consumers have a significant impact on food safety risk communication on social media. Overall, the protection motivation theory accounted for 71% of the variance in food safety risk communication on social media. Practical implications and suggestions are proposed for the related stakeholders, as well as consumers, to encourage the public to participate in the food risk communication in this study. The research findings presented the social media as a kind of food risk communication channel contributes to consumers acquire accurate information on food quickly, in turn, reduce the probability of food poisoning in daily life. Protection motivation theory may provide some insights into how we can increase the rate of food safety risk communication on social media.

## 1. Introduction

Suan Tang Zi are fermented noodles made from maize by the residents of northeast China; they are prevalent in eastern Liaoning, southeastern Jilin and eastern Heilongjiang. On 5 October 2020, a food poisoning incident occurred in Jidong County, Jixi City, Heilongjiang Province, caused by the consumption of Suan Tang Zi at a family dinner, in which all nine people died after eating them. According to a local police investigation, the cause of death was identified as the family members’ consumption of Suan Tang Zi; the homemade noodles were contaminated with germs after being frozen for almost a year, which led to the family members contracting Bongkrek acid (a respiratory toxin produced by bacteria found in food). The Suan Tang Zi accident shows that food safety issues should not be ignored, and consumers must attach great importance to the safety of homemade foods.

It is estimated that between 200,000 and 400,000 people are poisoned by food each year in China [1], accounting for between 0.014 percent and 0.03 percent of China’s total population. Therefore, food safety risks have become a significant threat to public health [2,3]. Globally, foodborne agents cause an estimated 600 million cases of illnesses and 420,000 deaths each year. Food poisoning or foodborne illness cannot be addressed by scientific advancements alone; behavior change at the individual level also plays a crucial role [4]. Food safety risk communication activities have been carried out in various provincial and municipal areas such as Jiangsu, Hunan, Kunming, and Shenzhen. However, food safety risk communication is still in its preliminary stages of exploration, mainly in the context of the actual food safety supervision at the grassroots level. In order to address the outstanding problems that currently exist in food safety, involving the definition, basic principles, classification, mechanism, content and system of food safety risk communication, we must enhance awareness on food risk communication of consumers.

Furthermore, China’s central government has implemented plenty of legislation in an attempt to strengthen food safety supervision and promote improved food safety standards. This includes the newly revised Food Safety Law of the People’s Republic of China, establishing a food safety risk communication system which marked the incorporation of risk communication into China’s legal system. Food safety risk communication is a particularly important part in the risk analysis process, which aiming to reduce the hazards caused by food safety risks for consumers. However, food risk communication is a complex activity involving many different communicators, including food experts, the media, the authorities, industry and consumers [5]. Media is an important medium for consumers to communicate about food safety risks. The 46th Statistical Report on Internet Development in China pointed out that as of June 2020, the number of Internet users reached China is 940 million people, of which the number of mobile phone users is as high as 932 million, and the Internet penetration rate is 67%. With the increase in the penetration rate of mobile and Internet users, and the frequency of using social media, it is necessary to understand the factors that affect consumers’ food safety risk communication through the medium of social media. 

Northeast China is located at a high latitude and experiences a long winter. Residents have developed the habit of storing homemade food for winter. The risk of homemade fermented food in northeast China has long been ignored by the residents of northeast China. However, in the absence of food preparation and storage knowledge, homemade food can easily cause food safety accidents. At this moment, food safety risk communication is particularly important. Although a number of studies have been conducted, current research on consumers’ protection motivation in food safety risk communication is still lacking, especially in comparison to risk perception of food safety. PMT integrates cognitive process with information, knowledge, attitudes towards actual behaviors [6], which apply to the context of this research. To date, most scholars have focused on design as an effective tool for food risk communication, or explored the factors that influence the communication effectiveness of consumers [7,8,9,10,11]. However, only a few studies have explored the behavior of food risk communication through the lens of protection motivation theory. Thus, to address this research gap, this study will focus on consumers’ food safety risk communication through social media, integrating the Suan Tang Zi accident. More specifically, we explore the relationship between cognitive mediating process, protection motivation and food safety risk communication on social media. It aims to promote food risk communication between stakeholders on social media. The results of this study will help to enhance consumers’ awareness of food safety risk communication, and at the same time arouse consumers’ attention to strengthen their knowledge of homemade food. As a result, it also shows that food safety risk communication not only requires government guidance, but also requires consumers to improve their subjective initiative in risk communication.

## 2. Literature Review

### 2.1. Food Safety Risk Communication

Risk communication is a fundamental part of risk analysis theory. Food safety risk communication is based on the exchange of information and views on the risks and risk-related factors associated with food safety hazards. It involves the communication of risks and benefits, i.e., providing information about risks and benefits of certain foods and enabling people to make rational decisions on food choices [12]. Food safety risk communication research has focused on conveying risk information, receiving risk information, and assessing the optimal medium for risk communication.

Food safety risk communication is an interactive, two-way exchange activity which includes a communicator, exchange information and an exchange platform; it also addresses the scope of exchange. It is an important way to convey food safety precautions to consumers, which can improve their knowledge around food and food-handling practices. As stakeholders, consumers are very important communication agents in food safety risk communication. Hence, most scholars have analyzed food safety risk communication frameworks and risk communication strategies around consumers. For instance, Cope et al. (2010) proposed a risk communication framework based on consumer preferences for different approaches to food risk management and noted that risk communication should be based on knowledge of consumer risk perceptions and information needs, including individual differences in consumer preferences and needs, as well as differences in socio-historical contexts related to regulation [13]. Cho et al. (2017) designed a three-step risk communication framework consisting of formative assessment, implementation, and outcome assessment through a group interview approach in the context of the Fukushima nuclear accident; the study focused on alleviating consumers’ concerns about radioactive contamination in food, which effectively increased consumers’ food knowledge and alleviated their anxieties around food safety [9].

In terms of receiving risk information, the impact of cultural differences, consumer knowledge and elements of consumer concern on the effectiveness of food safety risk communication is highlighted from the perspective of risk communication effectiveness and the development of consumer food safety risk communication strategies. Dijk et al. (2008) selected three types of food hazards in organic food: mycotoxins, pesticide residues and genetically modified potatoes; the study measured mixed linear models to compare the impact food risk information and related risk management practices on consumers’ perceptions of the quality of food risk management in different European countries, highlighting the importance of cultural differences in influencing potential risk communication strategies [14]. Christopher Griffith et al. (1998) used an observational approach to assess the risk of home-prepared food poisoning and showed that the majority of consumers failed to implement basic hygiene practices due to lack of basic knowledge or understanding of food safety protocols, resulting in food poisoning incidents [15]. Charlebois & Summan (2015) proposed a core-risk communication strategy based on a risk communication model for food safety supervision authority, and it stipulates communication strategy between the consumer and the food industry; they also proposed continuous evaluation and improvement processes [16]. Li et al. (2020) used factor analysis and cluster analysis to study the food risk communication of three types of parents in rural China: sensitive, dependent and conservative. The results showed that the effect of food risk communication depends on the elements which the interviewee cares about most; this opens an effective way for food risk communication in rural China [2]. McCarthy and Brennan (2009) investigated food risk communication in Ireland; they analyzed the barriers to effective communication including personal, infrastructural, and information related factors such as lack of interest and conflicting information [17]. Furthermore, the authors noted the role of the media in influencing public perceptions of risk, as well as providing space and opportunities for expert-public dialogue, and concluded with specific measures for effective food risk communication. It is necessary to adopt differentiated communication methods for consumer groups with different characteristics to increase the public’s confidence in obtaining food safety information sources. Consumer choice of food safety information sources depends on several elements, with different people relying on different sources [18]. Providing relevant risk information to vulnerable consumers and target groups requires an in-depth understanding of the recipients of the information [19]. Tiozzo et al. (2018) investigated Italian consumers’ sources of food safety information, noting that age, education, employment status, family status, and level of objective knowledge were influencing Italians’ choice of information sources. Their results indicated that Italian consumers care about food safety and actively seek information [20]. Crovato et al. (2016) used paired sample t-tests and ANOVA to study the effectiveness of a pilot health-related project conducted in Italy to raise risk awareness among adolescent consumers [8]. Liu et al. (2014) explored the patterns of access to food safety information by different segments of the population, suggest that strengthening cooperation with the government, doctors and research institutions can improve public confidence in the reliability of food safety information [21]. Furthermore, the internet as a channel for food risk communication can help the public to access credible information, and it should be selected according to the needs of the target consumers.

According to the summary of existing literature about food safety risk communication, it can be found that successful communication about food safety risk depends on reliable sources, clear and effective information, and focuses on the real needs and perceptions of the communicator [22]. Food safety risk communication is influenced by individual characteristics, the degree of public trust in the government and the information sources [18,23,24,25,26,27,28] and information seeking [11,13,28,29]. 

### 2.2. Communicate Food Safety Risk on Social Media

The public tends to rely on food related information not only from official sources, but also from their friends, peers, and family [30]. Social media as an important instrument for communicating instant information on food safety risks, which allows users to interact with message producers and each other. Compared to traditional media, the timeliness, interactivity, and free participation of social media has attracted many users, leading to a continuous growth in the number of people on social media sites. This provides a good opportunity for research on food safety risk communication. Social media offers all individuals the opportunity to spread information about the risks and benefits of food.

Despite the early popularity of social media, food safety risk communication has initially been a one-way communication. For example, Regan et al. (2016) conducted in-depth interviews with key stakeholders in risk management and communication in the Irish food industry and found that most stakeholders did not value two-way risk communication in food crises [31]. Investigating their willingness to adopt and effectively use social media, the results of the study showed that the stakeholders interviewed were aware of the need to engage with social media in times of food safety crises, but most regarded it as a one-way channel to help spread a particular message. Rutsaert et al. (2013) discussed the current status of social media and its potential as a tool for the communication of food risks and benefits [30]. 

Providing relevant risk information to target groups requires in-depth knowledge of the recipients of this information and social media is uniquely suited to the effectiveness of risk communication [19]. The advantages of social media in food safety risk communication have been studied. In the early stages of these studies, attention was paid to the relationship between social media and traditional media when reporting on food risk events. Results showed that traditional media relied on offline resources to report on various events, while social media was more responsive and reactive, though there was significant difference in the negative tone between the different types of media [32]. Li et al. (2020) noted that social media platforms such as Weibo and WeChat have been used as the main channels for rural residents to obtain information about food safety risks and to communicate about food safety issues [2]. Since then, the food safety risk information-seeking intention of consumers in WeChat, alongside other influencing factors, have been researched by Zhaohui Yang (2020) [33]. A strategy-oriented approach was used to investigate the views of food industry stakeholders and experts on the potential use of emerging media in communicating food risks benefits, with the results indicating that the role of social media in food safety risk communication is significant [34]. In addition, some scholars have found that factors such as trust, personal beliefs, risk perception, emotions, and social support are all significant factors that influence consumers’ willingness to use social media for food safety risk communication [7,35,36]. Literature which summarizes how social media can be better used for food risk communication also exists, noting that trust and personal beliefs are important drivers of social media use [4]. 

Based on the above literature review, the results of food risk communication and the use of social media for risk communication are fruitful and provide a basis for current research. However, due to poor consumer perception, lack of motivation and food expertise, ways to improve consumer risk communication remain the focus of food risk assessment experts. Firstly, the literature has examined the impact of the reliability of food risk information sources on consumer food safety risk communication, but has not considered consumers’ cognitive conditioning processes, i.e., the combination of threat appraisal and coping appraisal after consumers receive information, creating an intervening variable of protection motivation. Conservation motivation is similar to other types of motivation in that it motivates, sustains and directs activity [37]. Therefore, after receiving risk information, the process of cognitive conditioning of the information by the consumer should be considered in order to understand the consumer’s food safety risk communication through the framework of motivation. Secondly, most of the previous studies are based on risk analysis theory [15], which analyzes the food consumption behavior of consumers with different levels of subjective and objective knowledge and reveals that consumers’ subjective and objective knowledge, as well as their consumption behavior, influences the effectiveness of food risk communication. Consumers’ willingness to communicate food risks is motivated by their own protection, which is best measured by behavior as a mediating variable [38]. Thirdly, although there is literature on the development of a consumer food safety risk communication framework, there is no corresponding analysis when faced with a specific food risk event. Therefore, research into a particular poisoning incident can provide reliable data for research on food safety risk communication.

Many food safety risk communication studies do not provide empirical evidence of the effect of protection motivation theory on using social media, and fewer focus on specific food safety accidents [4,39]. Furthermore, PMT is considered one of the most powerful explanatory theories in predicting an individual’s intention to take protective measures. The appraisal process of the PMT contains the comparison of threats and benefits, and the balance of efficacy and costs, which are key to analyzing the actions of consumer food safety risk communication on social media. In order to address these research gaps, we integrate Suan Tang Zi accident to predict the behavior of consumer about food safety risk communication on social media with the protection motivation theory in this study. To achieve our objective, we conducted the structural equation modeling (SEM) process to evaluate the PMT model in Figure 1. The results may provide insightful empirical implications conducive to formulating an effective strategy for food safety risk communication on social media for related stakeholders. Furthermore, the research enriched the application of the extended PMT model in the field of food safety risk communication.

## 3. Theoretical Foundation and Hypotheses

### 3.1. Theoretical Foundation

Protection Motivation Theory (PMT), developed by Rogers (1983), describes the protection attitudes and behaviors of an individual who is exposed to a threat [40]. Drawing from expectancy-value theories, PMT explains the cognitive processes that individuals experience when faced with threats [41]. PMT posits that two underlying processes, threat appraisal and coping appraisal, underlie peoples’ adoption of protective behaviors when faced with a threat or hazard (see Figure 2). Threat appraisal is a process of estimating the severity and vulnerability of a threat, while coping appraisal refers to the process of evaluating the response efficacy and self-efficacy of the individual who is exposed to the threat [42].

Threat appraisal involves an individual’s assessment of the degree of risk which comes with the adverse consequences posed by a threatening event or unsafe behaviors [43,44]. Threat appraisal consists of two components. The first is perceived severity, which represents the severity of the consequences of anticipated threats. In this research, perceived severity is taken to measure the severity of threats caused by eating metamorphic-fermented food. The second is perceived vulnerability, which comprises of the assessment of the likelihood of threat events. In this research, perceived vulnerability is the probability of bad results which occur when homemade fermented food is eaten. 

Coping appraisal involves the evaluation of one’s capacity to deal with and avoid a threatening event [43,45]. It consists of three sub-constituents: response efficacy, self-efficacy, and response barriers. Response efficacy is related to one’s belief about the perceived benefits of establishing coping behaviors. Self-efficacy is the assessment of one’s perceived ability for adaptive behavior. Response barriers emphasize the potential costs of coping mechanisms such as time, effort, money, etc. In this research, the coping behavior is identified as communicating food safety risk on social media.

PMT has been widely used in the history of health research, environmental protection and related to human behaviors on computers [40,46,47,48,49,50,51]. In the context of food safety risk, due to the profit-driven nature of the food industry and the deficit knowledge of the consumer, a food hazard may be caused by an organization or an individual’s behavior, which can lead to food poisoning or threaten physical health. Motivated by self-protection, individuals tend to extend the boundaries of food safety knowledge. Therefore, PMT is suitable for the current study that aims to investigate food safety risk communication behavior, following the Suan Tang Zi incident in Heilongjiang Province, China. 

### 3.2. Research Hypotheses 

PMT can explain how and why people behave to protect their health (Floyd, Prentice-Dunn, & Rogers, 2000). Through the lens of PMT, this research develops a framework to investigate the consumers’ food risk communication behavioral intentions, as depicted in Figure 2. Threat appraisal and coping appraisal jointly affect one’s motivation for physical protection. One of the most notable focuses shared by food safety risk studies and PMT is consumers’ risk perception, including perceived severity and perceived vulnerability [52]. 

#### 3.2.1. Perceived Severity and Perceived Vulnerability 

According to PMT, maladaptive perception is an important construct that affects people’s personal health behaviors. Severity is “the degree of physical harm, psychological harm, social threats, economic harm, dangers to others rather than oneself, and even threats to other species” [53]. Perceived severity represents an individual’s assessment of the severity of the consequences resulting from threats [51,54,55]. The more seriously a person perceives the hazards of a catastrophic event or unsafe behavior, the more attention that person will pay to the event. Perceived vulnerability refers to the conditional probability that the threatening event will occur provided that no adaptive behavior is performed or there is no modification of an existing behavioral disposition [53].

Similarly, in this study, perceived severity refers to consumers’ perception of the implications of poor food handling methods, including the threat to their physical health caused by contaminated food, and the economic loss caused by lack of food risk communication. Perceived vulnerability refers to the likelihood that consumers perceive the threats of food safety risk. In the Suan Tang Zi accident, perceived severity refers to consumers’ perception of the extent of danger caused by the Suan Tang Zi accident. Perceived vulnerability indicates the conditional probability that the Suan Tang Zi threatening event will occur. Consumers’ perception of severity and vulnerability will be more serious, their fears will be aroused, and their protection motivation will be stronger. Thus, the hypotheses are as follows.

**Hypotheses** **(H1).**
*Perceived severity has a significant positive effect on the protection motivation of consumers.*


**Hypotheses** **(H2).**
*Perceived vulnerability has a significant positive effect on the protection motivation of consumers.*


#### 3.2.2. Response Efficacy and Self-Efficacy

Coping appraisal refers to an individual’s estimation of the degree of loss or damage that might be caused by a threatening event. It can also be understood as one’s ability to address and avoid threats. Response efficacy is the belief that an adaptive response will work and that taking protective measures will be effective in protecting oneself or others (Floyd, Prentice-Dunn & Rogers, 2000). Protection motivation theory explicitly refers to self-efficacy [56]. Self-efficacy is the perceived ability of an individual to actually execute the adaptive response; it also refers to the problems that individuals expect to encounter in adopting precautious methods or doubts about their ability to change their current patterns of behavior [57]. 

In the context of the food field, response efficacy refers to consumers’ perception of the effectiveness of their own adaptive behaviors. For instance, this refers to the extent to which consumers believe that the behavior of proactive communication about food safety risk is effective in maintaining their health. Generally, consumers take an action because they believe that they will benefit from it. Self-efficacy refers to consumers’ awareness of their ability to take the initiative to engage in food risk communication behavior. Moving on from the theory foundation, it is necessary to explore the relationship between the efficacy of food safety risk communication on social media and the protection motivation of consumers following the Suan Tang Zi accident. Therefore, the hypotheses are as follows.

**Hypotheses** **(H3).**
*Response efficacy has a significant positive influence on the protection motivation of consumers.*


**Hypotheses** **(H4).**
*Self-efficacy has a significant positive effect on the protection motivation of consumers.*


#### 3.2.3. Response Barriers

Response barriers refer to all the perceived costs associated with protective measures or actions, including monetary and non-monetary costs (e.g., effort, time, and inconvenience) (Yandong Wang et al., 2019). Food safety risk communication behavior is not only influenced by efficacy but also by response barriers. The barriers to effective food risk communication include personal, infrastructural and message-related factors, such as lack of interest, lack of appropriate facilities and conflicting messaging (McCarthy, Brennan, 2009).

In the food context, response barriers refer to the barriers that consumers encounter when performing food safety risk communication on social media, including financial costs, time costs and personal effort. Response barriers can reduce the extent to which consumers perform food safety risk communication. In addition to the above, there are also barriers to food safety risk communication on social media, such as the incomprehensible food terminology and the negative emotions felt by consumers when interacting with each other, particularly following the Suan Tang Zi accident. Therefore, the following hypothesis is formulated.

**Hypotheses** **(H5).**
*Response barriers have a significant negative influence on the protection motivation of consumers.*


#### 3.2.4. Protection Motivation and Food Safety Risk Communication Behaviors

Protection motivation is a mediating variable with typical characteristics of motivation, which causes, maintains, and guides activities. Protection motivation arises from the perception and evaluation of hazards and incidents and suggests that response measures can effectively prevent threats (Rogers, 1983). Behavior refers to a person’s engagement or expression [58]. 

In the context of food safety communication on social media, prior studies have shown that trust and personal beliefs drive media use (Overbey et al., 2017). In fact, an individual’s protection motivation stimulates the acceptance of their response [59]. In the context of this study, PMT indicates that motivated people are inclined to take protection measures (Reza, 2021). Following the Suan Tang Zi accident, consumers are becoming more vigilant about fermented foods. Strong protection motivations may reduce food safety risk through communication on social media. Accordingly, the following hypothesis is proposed.

**Hypotheses** **(H6).**
*Protection motivation has a significant positive effect on food risk communication on social media.*


#### 3.2.5. Information Need

Information needs present the gap between consumers believing they need sufficient information on food safety risks and their knowledge surrounding food safety risk. Food risk communication should be informed by knowledge of consumer risk perceptions and information needs (Cope, Frewer et al., 2010). Media plays an important role in the risk communication, especially on constructs of attitudes and beliefs [21,60,61]. Previous research shows that consumers rely on information provided by the mass media and accurate coverage on foodborne risks contribute to public receive education (Tiozzo et al., 2018). Considering the information needs of consumers can strengthen the effectiveness of food safety risk communication (Cope et al., 2010; Vainio et al., 2020). However, perceived risk influences information needs, information seeking and processing (Qiaozhe Guo et al., 2020). Consumers have a high demand for accurate information, but their searching behaviors are too passive to meet their information needs (Kim et al., 2015). The information reserves of each individual are different because of different educational backgrounds and personal ability. Information needs will eventually affect information seeking behavior [62]. Seeking information on food safety is a kind of food safety risk communication. Therefore, the following hypothesis is developed.

**Hypotheses** **(H7).**
*Information needs have a significant positive effect on food safety risk communication on social media by consumers.*


## 4. Research Methodology

### 4.1. Sample and Data Collection

To test the hypotheses presented in the above theoretical model, and establish the relationships between the variables, we collected data through an online survey. The questionnaire was administrated in Heilongjiang, Jilin, and Liaoning Province, which includes Jixi, Qitaihe, Peony River, Hegang, Shuangyashan, Harbin, Yanji, Changchun, Benxi, Dandong, Tieling city. We needed to measure the psychological variables of the threat appraisal and the coping appraisal. Questionnaires were sent to consumers by Sojump between March and April 2021. We collected 789 questionnaires, among which 113 had invalid responses. Thus, 676 questionnaires with valid responses were left. 

Before the formal survey, we conducted a three-step survey. Firstly, we discussed the questionnaire in the groups, and the members proposed suggestions. According to the suggestions, the questionnaire was revised. Secondly, we conducted pre-research online. Fifteen interviewees were randomly selected to participate in the questionnaire after it had been revised, allowing us to conduct a preliminary test (the answers were excluded from the final sample). The questionnaire was further improved based on their feedback, and some measurement items were appropriately adjusted to make the questionnaire easier to understand in the Chinese environment. Compared with field research, an online survey has advantages such as quick collection of results, highly controllability and the chance for respondents to provide more objective responses without disturbance. According to our research members, all respondents they examined were Suan Tang Zi consumers. We paid 6 CNY (about $0.9), through Sojump, to every respondent who filled out one questionnaire. At the top of the questionnaire, the concept of food safety risk communication was provided so that the participants could understand before filling out the questionnaire. Moreover, we explained the research purpose and promised that the data collected was only for academic research. Additionally, we explained that the respondents’ information would be strictly confidential and their personal information would never be leaked, allowing the participant to answer with confidence. 

### 4.2. Measurement

The questionnaire included thirty-three items on eight different constructs and was divided into six parts. Most of the items in our measurement model were adopted from existing studies. The first part was the threat appraisal of Suan Tang Zi on respondents. The second part was the coping appraisal of Suan Tang Zi on respondents. The third part was the protection motivation of respondents. The fourth part was the information needs of respondents on food risk. The fifth part was the food safety risk communication of the respondents. The sixth part was demographic characteristics, including basic information such as gender, age, marital status, education level, and income of the respondent. Our analysis has used only some parts of original questionnaire by using protection motivation of respondents to predict the food safety risk communication. Other information acquired will be used in another research. All of the items were responded to on a five-point Likert scale ranging from 1 (strongly disagree) to 5 (strongly agree). Constructs and measurement items are shown in Table 1. All of the measurement items were adapted from existing literature [11,51,54,55,63,64,65,66,67,68].

We used structural equation modeling (SEM) to test our model. SEM is a flexible and powerful extension of the general linear model. The rationality of the causal model is tested by SEM. Thomopson (2004) proposed the analysis of the measurement model before analyzing the structural model. If the fit index of the measurement model is acceptable, one can perform a complete SEM model evaluation. CFA is a confirmatory factor analysis for each latent variable to determine whether the hypothetical measurement model is satisfactory. We used the analysis of the moment structures (AMOS24.0) program to estimate the parameters of this study. Model adjustment was analyzed according to Barbara (2009) [69]. The Comparative Fit index (CFI) was 0.935, which was above the standard value of 0.9. Root Mean Square Error of Approximation (RMSEA) was 0.057. Goodness of Fit Index (GFI) was 0.901. χ^2^/df is 4.048. Thus, the structural model yielded a good fit. 

## 5. Results

### 5.1. Measurement Model

Table 2 reports the demographic characteristics of the respondents. For the gender scale, 32.1% of the respondents are male and 67.9% are female. In terms of age, the proportion of the respondents under 29 years old is 44.7%, and the remaining 55.3% are older than 29 years of age. 59% respondents are married. Respondents with bachelor’s degree or higher occupied 59.2%. More than 70% of respondents have an annual income below or equal to 70,000 CNY (about $10,787).

To assess the measurement model, we need to evaluate reliability, convergent validity, and discriminant validity. As shown in Table 3, the Cronbach’s alpha coefficient was calculated, with a value above 0.70 indicating good internal consistency reliability (Reza Mousavi et al., 2021). Hence, all constructs had an acceptable reliability. Composite reliability, factor loadings and average variance extracted (AVE) are used to examine the convergent validity of the measurement model. In the CFA, composite reliability ranged from 0.853 to 0.957, which is greater than the 0.7 benchmark value [70]. AVE values ranged from 0.662 to 0.882, which are above the benchmark value of 0.7 (Fornell & Larcker, 1981). Most factor loadings are above threshold value of 0.7 in this study. Lastly, we delete the items which do not satisfy the threshold value of 0.7. Obviously, all above criteria are satisfied. Therefore, our measurement model has good convergent validity. For discriminant validity, the three criteria are as follows: (a) the square root of AVE should be greater than the correlation coefficients between the particular construct and other constructs [71], (b) the loading of an item on its respective construct should be significantly higher than the loadings on other constructs [72]. As shown in Table 4, the correlation coefficients of latent variables and discriminant validity satisfy the above criteria. Therefore, the measurement model has good convergent validity.

### 5.2. Structural Model

The results of hypotheses tests in the Structural Equation Modeling are shown in Table 5. The significance of structural paths and the R^2^ of endogenous variables are usually used to evaluate the explanatory ability of the structural model. Figure 3 presents the estimated parameters, including path coefficients, significance level, and explained variances. From the results depicted in Figure 3, we can see that the proposed model explained 53% of the variance in protection motivation and 71% of the variance in food safety risk communication on social media. The standardized path coefficient between perceived severity of the Suan Tang Zi accident and protection motivation of the consumer is 0.08 (H1; b = 0.08, *p* < 0.1), which indicates that perceived severity has positive effect on protection motivation of the consumer. Perceived vulnerability has a significant positive effect on protection motivation of the consumer (H2; b = 0.16, *p* < 0.01). Self-efficacy has a significant positive effect on protection motivation of consumer (H4; b = 0.65, *p* < 0.01). Response efficacy has a positive influence on protection motivation of the consumer (H3; b = 0.05, *p* = 0.381). Response barriers have a negative influence on protection motivation of the consumer (H5; b = −0.02, *p* = 0.413). Protection motivation of the consumer has a significant positive influence on food safety risk communication on social media (H6; b = 0.27, *p* < 0.01). Information needs of the consumer have a significant positive effect on food safety risk communication on social media (H7; b = 0.67, *p* < 0.01). Therefore, the assumptions H1, H3 and H5 are not supported. Additionally, Figure 3 shows no significant relationship between response efficacy, response barriers and protection motivation of the consumer. Thus, H1, H3 and H5 are not significant. 

Furthermore, we considered the indirect effects of the model through bootstrapping (Taylor, MacKinnon, et. al, 2008) [73]. Bootstrapping is already used widely in SEM to test indirect effects. In Table 6, we performed direct, indirect, and total effects at a 95% confidence interval with 1000 bootstrap samples. We referenced the views of Preacher and Hayes (2008) and used the confidence interval of the lower and upper bounds to test whether the indirect effects existed [74]. If zero is not between the confidence interval, then we claim that the indirect effects exist. As shown in Table 6, the results indicated that protection motivation only mediates the effect of self-efficacy on food risk communication (indirect effect = 0.168). 

## 6. Discussion

This study explores the factors which influence food safety risk communication of consumers using social media. It emphasizes perceived severity, perceived vulnerability, response efficacy, self-efficacy, and response barriers in food safety risk communication on social media. Importantly, this study is the first to investigate all of the PMT constructs in the context of food safety risk communication. 

The conclusions build on and are broadly consistent with prior research [25,28,47,75]. Our study makes several contributions to theory building. First, the PMT was used to analyze the influencing factors of consumers’ food safety risk communication behavior on social media, which has contributed to the food safety risk communication literature. In previous studies, no attention was paid to the relationship between PMT and food safety risk communication. Our work provides evidence that perceived severity, perceived vulnerability, and self-efficacy have a significant positive influence on protection motivation of the consumer in food safety risk communication. The protection motivation has a significant positive influence on food safety risk communication on social media. Additionally, we find that the information needs of the consumer have a significant positive influence on food safety risk communication on social media. However, the communicators easily ignore the information needs of the consumer. Thus, we should pay more attention to information needs of consumer in food safety risk communication. Secondly, this research extends the application of PMT in the field of food safety risk analysis, sheds light on PMT, and highlights the relationship between consumers’ information needs and food safety risk communication behavior. 

Moreover, threat appraisal and coping appraisal also have a significant influence on protection motivation in other fields (Reza Mousavi et al., 2021; Mei-Fang Chen, 2020). However, response efficacy and response barriers have a limited impact on consumer protection motivation in food safety risk communication.

## 7. Conclusions

The Suan Tang Zi accident was caused by the improper behavior of consumers and a lack of knowledge surrounding homemade fermented food. This study examined the food safety risk communication behavior of consumers in northeast China. Firstly, PMT was introduced to extend the current exploration factors on how to affect food risk communication behavior of consumers on social media, following the Suan Tang Zi accident. Our results indicated that protection motivation and information needs have a significant positive effect on food safety risk communication, with information needs being the more significant influencing factor of the two. Perceived severity, perceived vulnerability and self-efficacy have direct effect on protection motivation. Response efficacy has a positive impact on protection motivation, and response barriers have negative impact on protection motivation, but neither were significant. Therefore, with regard to the actual food safety risk environment, perceived severity, perceived vulnerability, and self-efficacy were the key determinants in influencing the protection motivation of consumers. Notably, protection motivation and information need have direct impact on food safety risk communication of consumers. The significant impact of information needs on food safety risk communication also indicates that consumers have a large knowledge deficit in terms of food safety risks, especially when it comes to of homemade fermented foods. 

Secondly, compared with other traditional communication routines regarding food safety, social media is more convenient and efficient. There is a practical significance to exploring the factors that influence consumers’ food risk communication behavior on social media. People are seldom cook during working day, and their knowledge of food preparation is relatively scarce, especially for young people. Therefore, it is necessary to enhance the frequency of food safety risk communication across different channels to reduce the likelihood and frequency of food poisoning events, especially in light of the Suan Tang Zi accident. The government needs to make an effort to promote food safety risk communication. Government guidance and interference can contribute to avoid the spread of fake news. It can also convey different types of food safety risk knowledge to the target audience. 

Thirdly, it is notable that a person’s beliefs about his or her ability to successfully perform particular preventive behaviors have a significant impact on protection motivation when faced with food risks. Thus, we should simplify professional food safety knowledge for consumers with low levels of education, and food safety risk communication should be carried out in a more intuitive and easy-to-understand manner. Food risk communication should be carried out in conjunction with public health and food safety education programs. Finally, we should pay more attention to the potential food safety risks that may exist in regional food cultures, such as homemade foods such as fermented soya-bean milker, which is popular in Beijing. While protecting the traditional food dietary culture, it is necessary to use social media to popularize the knowledge of homemade food and storage practices.

## Figures and Tables

**Figure 1 ijerph-18-08080-f001:**
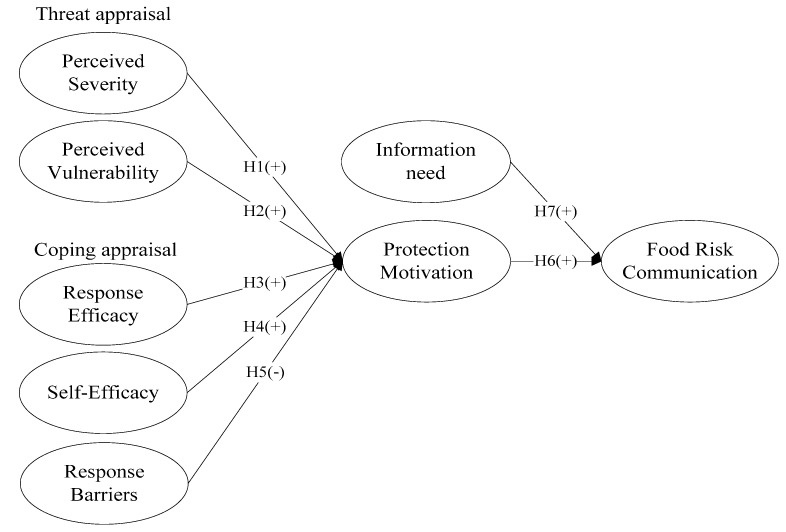
Research model.

**Figure 2 ijerph-18-08080-f002:**
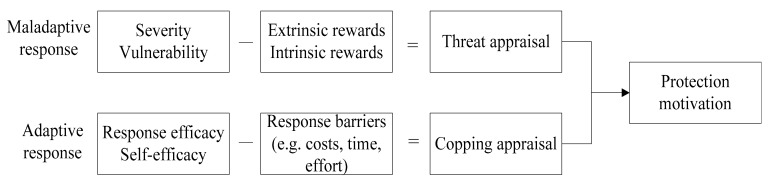
Protection motivation theory. Source: Rogers, 1983.

**Figure 3 ijerph-18-08080-f003:**
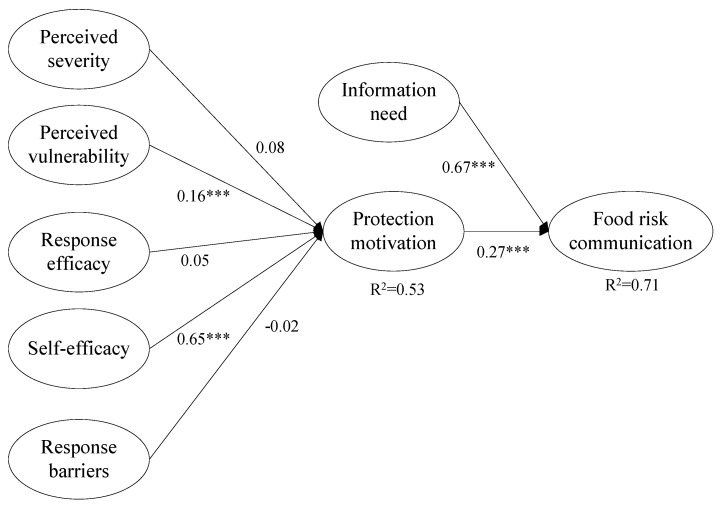
Results of the structural model analysis. Notes: *** *p* < 0.01 indicated that *p*-values are significant.

**Table 1 ijerph-18-08080-t001:** Constructs and measurement items.

Constructs	Measurement Items	Mean	S.D.
Perceived severity (PS)	PS1. I realized homemade fermented food is a serious food safety issue after Suan Tang Zi accident.	4.26	0.936
	PS2. I realized eating contaminated food is harmful to me after Suan Tang Zi accident.	4.43	0.845
	PS3. Homemade fermented food is more deadly than most people realize.	4.35	0.873
Perceived vulnerability (PV)	PV2. Homemade fermented food such as Suan tang Zi may threaten my physical.	3.99	0.925
	PV3. Contaminated food is highly likely to cause a significant harmful to our physical.	4.30	0.814
	PV4. I am vulnerable to harming by eating contaminated food.	4.20	0.838
Response efficacy (RE)	RE2. Communicating food safety risks on social media would stop me from foodborne illness.	4.13	0.824
	RE3. Communicating food safety risks on social media to make sure that I avoid from harmful with food safety.	3.99	0.915
	RE4. Food safety risk communication on social media help to reduce the risk of property damage caused by food safety.	4.14	0.795
	RE5. Food safety risk communication on social media help to reduce the risk of lives caused by food safety.	4.15	0.780
	RE6. It is effective to obtain knowledge of food storage, food handling practices, food quality identification, dietary balance and nutritional formula from social media.	4.13	0.793
Self-efficacy (SE)	SE1. I know how to communicate food safety risk effectively on social media to reduce my risk of food issues.	4.05	0.804
	SE2. I am able to communicate food safety risk on social media when I want to.	4.04	0.829
	SE4. I have confidence with the information of food safety risk prevention measures published by food experts or doctors on social media.	4.07	0.807
	SE5. I am confident that I can protect myself against food safety risk by communicating food safety risk on social media.	4.04	0.805
	SE6. I would be very interested if the food safety news on social media is closely related to my diet.	4.14	0.765
Response barriers (RB)	RB1. Communicating food safety risk on social media is time consuming.	3.23	1.206
	RB2. Communicating food safety risk on social media need cellular data consuming and increase my cost.	3.11	1.238
	RB3. Communicating food safety risk on social media make me displeased.	3.07	1.263
	RB4. The professional terms of food safety on social media make me incomprehensive.	3.25	1.187
Protection motivation (PM)	PM1. I would like to get more food safety knowledge on social media to Figureht the food risk.	4.11	0.776
	PM2. I would like to involve the food safety supervision work with others on social media to protect the food safety.	4.08	0.762
	PM3. I would like to involve the food safety propaganda on social media to improve the coping strategy of public about food risk.	4.12	0.775
	PM4. I would like to consult the professional through social media about homemade fermented food like Suan Tang Zi to protect my family.	4.11	0.763
	PM5. I would like to search food safety news through social media to protect me from being harmed by contaminated food.	4.11	0.769
Information need (IN)	IN1. I need more information related to homemade fermented food from social media.	4.05	0.769
	IN2. I would like to know more information about the homemade fermented food from social media.	4.07	0.758
	IN3. I need food safety information released by government through multiple channels especially official government account on social media.	4.15	0.742
Food safety risk communication on social media (FC)	FC1. I take the initiative to share food safety risk communication with friends and relatives on social media.	4.00	0.805
	FC2. I take the initiative to follow news about food safety issues on social media and communicate with others.	4.00	0.787
	FC3. I actively follow the food news published on the official Weibo/WeChat official account of the State Administration for Market Regulation and communicating with others.	4.02	0.795
	FC4. I take the initiative to evaluate and screen the authenticity of food information delivered on social media.	4.04	0.789
	FC5. I consult the professionals on food nutrition or food safety issues through social media.	3.99	0.829

Note: S.D. is standard deviation.

**Table 2 ijerph-18-08080-t002:** Demographic characteristics of the sample (*n* = 676).

Demographic Variable	Types	Frequency	Percentage (%)
Gender	Male	217	32.1
	Female	459	67.9
Age	<18	10	1.5
	18–29	292	43.2
	30–49	295	43.6
	50–59	61	9.0
	>60	18	2.7
Marital status	Married	399	59.0
	Single	277	41.0
Education level	Primary school	11	1.6
	High school	73	10.8
	Junior college	192	28.4
	Bachelor degree	284	42.0
	Postgraduate	116	17.2
Personal income (yearly)	Less than ¥30,000 ($4623)	263	38.9
	¥30,000–¥70,000 ($4623–$10,787)	242	35.8
	¥70,000–¥120,000 ($10,787–$18,492)	119	17.6
	¥120,000–¥200,000 ($18,492–$30,820)	41	6.1
	More than ¥200,000 ($30,820)	11	1.6

**Table 3 ijerph-18-08080-t003:** Confirmatory factor analysis results for measurement model.

Constructs	Items	Estimate	S.E.	C.R.	Factor Loading	CR	AVE	Cronbach’s *α*
PS	PS1	1			0.834			
	PS2	0.922	0.035	26.091	0.852	0.899	0.747	0.898
	PS3	1.011	0.037	27.341	0.906			
PV	PV2	1			0.706			
	PV3	1.007	0.053	19.102	0.808	0.853	0.662	0.848
	PV4	1.172	0.061	19.086	0.913			
RE	RE2	1			0.835			
	RE3	1.121	0.041	27.187	0.843			
	RE4	1.034	0.035	29.391	0.896	0.934	0.740	0.932
	RE5	0.994	0.034	28.997	0.877			
	RE6	0.977	0.037	26.738	0.848			
SE	SE1	1			0.787			
	SE2	0.926	0.037	24.819	0.707			
	SE4	1.221	0.052	23.290	0.868	0.916	0.688	0.915
	SE5	1.161	0.050	23.246	0.910			
	SE6	1.095	0.045	24.451	0.861			
RB	RB1	1			0.925			
	RB2	1.042	0.023	45.766	0.940			
	RB3	1.076	0.023	47.656	0.951	0.961	0.860	0.961
	RB4	0.949	0.024	38.725	0.892			
PM	PM1	1			0.866			
	PM2	1.038	0.031	33.972	0.915			
	PM3	0.989	0.033	30.381	0.858	0.949	0.788	0.947
	PM4	1.006	0.032	31.738	0.887			
	PM5	1.041	0.030	34.228	0.910			
IN	IN1	1			0.916			
	IN2	0.989	0.029	34.549	0.920	0.915	0.782	0.913
	IN3	0.854	0.030	28.382	0.812			
FC	FC1	1			0.852			
	FC2	0.989	0.024	40.823	0.861			
	FC3	1.054	0.033	31.768	0.909	0.943	0.767	0.946
	FC4	0.997	0.034	29.211	0.866			
	FC5	1.075	0.035	30.570	0.889			

Note: CR means composite reliability. AVE means average variance extracted.

**Table 4 ijerph-18-08080-t004:** The correlation coefficients of latent variables and discriminant validity. Note: The bold values in diagonal represent the sqrt (AVE) values.

Latent Variables	PS	PV	RE	SE	RB	PM	IN	FC
PS	0.864							
PV	0.678	0.814						
RE	0.518	0.655	0.860					
SE	0.495	0.616	0.843	0.829				
RB	−0.034	0.031	0.122	0.180	0.927			
PM	0.483	0.558	0.672	0.734	0.089	0.888		
IN	0.469	0.612	0.671	0.755	0.110	0.813	0.884	
FC	0.449	0.571	0.683	0.767	0.138	0.760	0.851	0.876

**Table 5 ijerph-18-08080-t005:** Results of hypotheses tests in the Structural Equation Modeling.

Hypothesis	Standardized Coefficient	S.E.	T-Value	*p*-Value	Decision
H1: PS->PM	0.08	0.036	1.677	0.093	Rejected
H2: PV->PM	0.16	0.046	3.279	0.001	Accepted
H3: RE->PM	0.05	0.058	0.875	0.381	Rejected
H4: SE->PM	0.65	0.062	9.994	***	Accepted
H5: RB->PM	−0.02	0.017	−0.818	0.413	Rejected
H6: PM->FC	0.27	0.034	8.039	***	Accepted
H7: IN->FC	0.67	0.038	17.767	***	Accepted

Note: *** *p* < 0.001.

**Table 6 ijerph-18-08080-t006:** Direct, indirect, and total effects of the hypothesized model.

Pathways	Point Estimate	Product of Coefficients		Bootstrapping			
				Percentitle 95%CI		Bias-Corrected 95%CI	
		SE	Z	Lower	Upper	Lower	Upper
*Direct effects*							
PM->FC	0.271	0.068	3.985	0.144	0.415	0.148	0.416
PS->PM	0.061	0.057	1.070	−0.044	0.141	−0.043	0.142
PV->PM	0.151	0.086	1.756	0.032	0.317	0.031	0.315
RE->PM	0.051	0.136	0.375	−0.256	0.263	−0.227	0.270
SE->PM	0.620	0.145	4.276	0.390	0.947	0.395	0.969
RB->PM	−0.014	0.019	−0.737	−0.050	0.023	−0.048	0.025
*Indirect effects*							
PS->PM->FC	0.017	0.017	1.000	−0.013	0.043	−0.008	0.046
PV->PM->FC	0.041	0.027	1.519	0.008	0.098	0.008	0.097
RE->PM->FC	0.014	0.039	0.359	−0.073	0.074	−0.070	0.076
SE->PM->FC	0.168	0.062	2.710 ***	0.078	0.311	0.081	0.319
RB->PM->FC	−0.004	0.005	−0.800	−0.015	0.006	−0.015	0.006
*Total effects*							
PS->FC	0.017	0.017	1.000	−0.013	0.043	−0.008	0.046
PV->FC	0.041	0.027	1.519	0.008	0.098	0.008	0.097
RE->FC	0.014	0.039	0.359	−0.073	0.074	−0.070	0.076
SE->FC	0.168	0.062	2.710	0.078	0.311	0.081	0.319
RB->FC	−0.004	0.005	−0.800	−0.015	0.006	−0.015	0.006
PM->FC	0.271	0.068	3.985	0.144	0.415	0.148	0.416
PS->PM	0.061	0.057	1.070	−0.044	0.141	−0.043	0.142
PV->PM	0.151	0.086	1.756	0.032	0.317	0.031	0.315
RE->PM	0.051	0.136	0.375	−0.256	0.263	−0.227	0.270
SE->PM	0.620	0.145	4.276	0.390	0.947	0.395	0.969
RB->PM	−0.014	0.019	−0.737	−0.050	0.023	−0.048	0.025

Note: 1000 bootstrap samples. *** *p* < 0.01.

## Data Availability

Data are available only upon request to the authors, according to the ethical approval from the Academic Committee of South China Agricultural University.

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
