# Peer review of "Consumers’ Food Safety Risk Communication on Social Media Following the Suan Tang Zi Accident: An Extended Protection Motivation Theory Perspective"

_ijerph, 2021, doi:10.3390/ijerph18158080_

Round 1
Reviewer 1 Report
The paper titled: “Consumers' food safety risk communication on social media following the Suan Tang Zi accident: An extended protection motivation theory perspective” is an interesting study on the important topic of human safety related to their basic living need, which is food.
The article is not easy to read, but it is linguistically correct.
In general, I consider the article very well prepared, especially the research part. The article presents the research problem logically and accurately, both from the literature and from examples of case studies. Very well discussed hypotheses and a thorough explanation of the research process. The procedure is transparent and explains the results obtained.
- My doubts are only raised by the introduction in which the authors try to emphasize the importance of the food safety problem, especially the numbers concerning deaths and diseases caused by the food problem given in absolute values do not show the scale of the problem. Perhaps it is worth presenting these food accidents as a share of the population.
- The research gap was somewhat poorly justified.
Questions to the authors: Is the obtained research sample representative? The internet survey strongly influences the suitability of young people with a higher status. Are these people who are at risk of a lack of knowledge and information about the dangers of improper food preparation or storage?
Author Response
Dear Reviewer, Thank you providing these insights. Appended to this letter is our point-by-point response to the comments.
Point 1: I My doubts are only raised by the introduction in which the authors try to emphasize the importance of the food safety problem, especially the numbers concerning deaths and diseases caused by the food problem given in absolute values do not show the scale of the problem. Perhaps it is worth presenting these food accidents as a share of the population.
Response 1: We agree with your assessment. Thus, we have increased the proportion of population who died from food poisoning in China. (Page2,lines 46-47)
Point 2: The research gap was somewhat poorly justified.
Response 2: We have added a more detail description of the research gap. (Page 2, lines 83-87)
Point 3: Is the obtained research sample representative? The internet survey strongly influences the suitability of young people with a higher status. Are these people who are at risk of a lack of knowledge and information about the dangers of improper food preparation or storage?
Response 3: Our research area covers the three northeastern provinces of Jilin, Liaoning and Heilongjiang. Most of respondents are above than 30 years old, and they have consumed Suan Tang Zi. In addition, the research samples are representative in terms of sample size. We agree with your suggestion that young people are more susceptible to internet influence than old people. Contemporary Chinese young people seldom cook during the working day, and their knowledge of food preparation is relatively scarce. Therefore, young people are also vulnerable to food risks. We proposed that it is necessary to strength the communication about food risks for young people (Page 16-17, lines 636-638, 652-663).
We sincerely appreciate you for your creative comments to improving the quality of this study.

Reviewer 2 Report
The manuscript is first of all very innovative. The work is very well documented (bibliography). Research hypothesis and methods are also good. The results correspond with the methods, and contribute important matters. As also wrote it is necessary to enhance the frequency of food safety risk communication with different channel for reduce the food poisoning
Author Response
Dear Reviewer,
Thank you very much for taking your time to review this manuscript. We really appreciate all your comments and providing these insights.
Reviewer 3 Report
The study is a well structured research with significant contributions to the literature. I have the following suggestions to improve the work.
It would be better if you can present your hypotheses test results in a table, showing the beta coefficients, p values, t values.
One or two variables have CR values above 0.95 which indicates redundancy.
Were the indirect effects considered for the study? If not, please do add a table showing the mediation effect results.
The conclusion looks too long, it would be better to add only the most important points to conclude the study.
Author Response
Dear Reviewer, Thank you providing these insights. Appended to this letter is our point-by-point response to the comments.
Point 1: It would be better if you can present your hypotheses test results in a table, showing the beta coefficients, p values, t values.
Response 1: We showed the bata coefficients, p-value, t-value, and hypotheses test result in table 5. (Page14,lines 546)
Point 2: One or two variables have CR values above 0.95 which indicates redundancy.
Response 2: You have raised an important point. However, it is acceptable for a few variables to have CR values above 0.95. We referenced “Xiaoxia Q et al. (2021) Development and validation of an instrument to measure beliefs in physical activity among (pre)frail order adults: An integration of the health belief model and the theory of planned behavior. Patient Education and Counseling” https://doi.org/10.1016/j.pec.2021.03.009
Point 3: Were the indirect effects considered for the study? If not, please do add a table showing the mediation effect results.
Response 3: According to your suggestions, we have added Table 6 to show the mediation effect results with bootstrapping method. (Page14,lines547)
Point 4: The conclusion looks too long, it would be better to add only the most important points to conclude the study.
Response 4: We have redrafted the conclusion to show the most important points in this study.
(Page 16-17, lines 636-638, 652-661).
We sincerely appreciate you for your creative comments to improving the quality of this study.
